# Immediate Effects of Wearing an Ankle Bandage on Fine Coordination, Proprioception, Balance and Gait in the Subacute Phase of Ankle Sprains

**DOI:** 10.3390/life14070810

**Published:** 2024-06-26

**Authors:** Tobias Heß, Thomas L. Milani, Anica Kilper, Christian Mitschke

**Affiliations:** 1Department of Human Locomotion, Chemnitz University of Technology, 09126 Chemnitz, Germany; 2Medical Center of Chemnitz, Department of Orthopaedics, Trauma and Hand Surgery, 09116 Chemnitz, Germany

**Keywords:** acute ankle sprain, ankle joint, ankle bandage, soft ankle orthoses, fine coordination and proprioception, single leg stance, Y-Balance test, modified Star Excursion Balance test, gait, rehabilitation

## Abstract

Ankle sprains are the most frequently occurring musculoskeletal injuries among recreational athletes. Ankle support through bandages following the initial orthotic treatment might be beneficial for rehabilitation purposes. However, the literature is sparse regarding the use of an ankle support directly after the acute phase of an ankle sprain. Therefore, this study investigates the hypothesis that wearing an ankle bandage immediately after an acute ankle sprain improves motor performance, stability and reduces pain. In total, 70 subjects with acute unilateral supination trauma were tested. Subjects were tested five weeks post-injury to assess immediate effects of the ankle bandage. On the testing day, subjects completed rating questionnaires and underwent comprehensive biomechanical assessments. Biomechanical investigations included fine coordination and proprioception tests, single leg stances, the Y-Balance test, and gait analysis. All biomechanical investigations were conducted for the subject’s injured leg with and without a bandage (MalleoTrain^®^ Bauerfeind AG, Zeulenroda-Triebes, Germany) and the healthy leg. Results indicated moderate to strong improvements in ankle stability and pain relief while wearing the bandage. Wearing the bandage significantly normalized single leg stance performance (*p* < 0.001), stance phase duration (*p* < 0.001), and vertical ground reaction forces during walking (*p* < 0.05). However, the bandage did not have a clear effect on fine coordination and proprioception. The findings of our study suggest that ankle bandages may play a crucial role in early-stage rehabilitation by enhancing motor performance and reducing pain.

## 1. Introduction

The ankle is a complex joint, with three degrees of freedom, that enables the body to adapt to different surfaces during physical activities and to absorb shocks and forces. During running and jumping, the forces exerted on the ankle joint can exceed several times the individual’s body weight [1,2]. Consequently, the ankle joint is one of the most stressed joints in the body, which can cause pain from injuries, such as fractures or sprains [3,4,5]. Ankle sprains occur mostly following uncontrolled, sudden movements, such as plantarflexion and inversion, which can lead to excessive force or strain on the ligaments of the joint. As a consequence, various ligamentous structures can be damaged, whereas the intensity of the trauma determines whether it results in a strain or a rupture [4,6,7,8,9]. Depending on the injured ligaments involved, three types of ankle sprains can be distinguished: medial, lateral, and syndesmotic [9,10,11]. Those types of ankle sprains can be classified by severity and duration. Severity is ranked in three grades based on the extent of tissue damage, as assessed through radiological and clinical examinations. Although there are different classifications, one classification is according to duration and is categorized as acute (up to 4 days post-injury), subacute (1 to 8 weeks post-injury), and chronic (more than 8 weeks post-injury). However, the exact timing for developing chronic ankle instability (CAI) is still debated [9,12,13]. Nevertheless, sprains do not only affect components of the musculoskeletal system, but also damage various components of the proprioceptive system [3,9,14,15,16,17,18,19]. This includes damage of specialized receptors, which are located in muscles, tendons, ligaments, and the joint capsule, such as Golgi tendon organs, muscle spindles, joint receptors and various mechanoreceptors. As these receptors provide important information about muscle length, muscle contraction speed, muscle tension, and joint position, for planning, adapting and executing movements, any disruption may negatively affect motor control [15,16,17,20,21,22,23,24].

From an epidemiological point of view, ankle sprains are the predominant type of injury among recreational athletes in various sports, accounting for approximately 49% of all injuries [5,6]. This includes non-contact sports, such as volleyball and running, as well as contact sports, such as basketball, handball, and soccer [5,9]. The lateral ligamentous structures of the ankle joint are mostly affected, which account for approx. 85% of ankle injuries [9,10,11,13,25,26,27]. Due to its high incidence, ankle sprains account for at least 14% of all emergency hospital visits, and is potentially even higher considering that 50% of individuals with ankle injuries do not report or seek hospital treatment [4,7,28,29,30]. Study results underline the excessive incidence of ankle sprains and their high impact on the economic and the healthcare system. Therefore, effective treatment, preferably in the acute and subacute phase, seems crucial.

The standardized treatment algorithm for high-grade (grade 2 & 3) ankle sprains call for all-day orthotic treatment for at least 5 weeks accompanied by active and passive physiotherapy [10,11,27,31]. The orthosis should be semi-rigid and stabilize the ankle joint. Thus providing mechanical support and controlling the range of motion (ROM) of the ankle joint [32,33,34]. Moreover, it enhances proprioceptive acuity, by stimulating cutaneous mechanoreceptors and joint receptors by compressing the underlying musculoskeletal structures [14,35,36]. With inadequate therapeutic support, residual symptoms, including pain, giving way, and impaired proprioception and neuromuscular control, can lead to chronic ankle instability (CAI) in 40 to 75% of individuals with ankle sprains [3,8,37,38,39,40]. In the worst-case scenario, CAI increases the risk of articular damage and the development of osteoarthritis. Additionally, it leads to substantial therapy costs and has a dramatic impact on patients quality of life [41,42].

An additional bandage treatment following the 5-week orthotic therapy can inform early intervention strategies, potentially reducing the incidence of CAI and improving patient outcomes. Unlike previous studies that have focused on the effects of bandages in healthy subjects or those with CAI, our study examines their immediate impact during the acute and subacute phase of injury. Therefore, the aim of our study was to investigate the immediate effects of wearing an ankle bandage on fine coordination, proprioception, and motor performance five weeks after orthotic treatment in patients with lateral ankle sprains. We hypothesize that wearing an ankle bandage immediately after an acute lateral ankle sprain will reduce pain, enhance fine coordination and proprioception, and improve motor performance, including balance and gait.

## 2. Materials and Methods

### 2.1. Participants

In total, 70 subjects with a subacute ankle sprain, caused by unilateral supination trauma were recruited for this interventional study. Subjects were recruited independently of their ethnicity and the cause of injury. Inclusion criteria were adults aged 18–60 years presenting with an acute unilateral supination ankle sprain (minimum grade 2) within three days post-injury. Exclusion criteria included grade 1 ankle sprains, upper leg sprains within the last 12 months, acute concomitant osseous injuries, history of confirmed lateral ligament injury, chronic ankle instability (CAI), neurological dysfunctions, rheumatic diseases, gout, arthrosis, recent surgeries, use of anticoagulants or corticosteroids, and any other conditions affecting motor performance or proprioception. If the patients were still suffering from the injury two to three weeks after the trauma, they were included in the study on a voluntary basis. After the initial treatment with orthotic supply according to the guidelines [10,11], the follow-up examination was performed approximately five weeks post-injury. This involved extensive clinical examination of the ankle joint (swelling, tenderness over the lateral ligament complex, and pain perception), followed by questionnaires and biomechanical investigations. Prior to the examination, all subjects were informed about the purpose of this study and provided written informed consent. All procedures were conducted according to the recommendations of the Declaration of Helsinki and were approved by the Ethics Committee of the Faculty of Behavioural and Social Sciences of Chemnitz University of Technology (V-320-17-LN-MalleoTrain-07042019).

### 2.2. Experimental Setup and Data Acquisition

All biomechanical tests were conducted on both the injured and healthy legs in a randomized order to minimize the influence of fatigue and habituation. Subjects performed familiarization trials with the ankle bandage and tasks before the main trials. All tests were carried out barefoot. The MalleoTrain^®^ bandage (Bauerfeind AG, Zeulenroda-Triebes, Germany) used in this study is made of an elastic, tight-fitting high-low knit material, providing alternating pressure massage during movement. Therefore, it exerts an alternating pressure massage during moving.

#### 2.2.1. Subjective Ratings

Patients provided anthropometric data and rated their pain when walking and standing barefoot, in shoes, with and without an ankle bandage using a visual analogue scala (0 to 10). Furthermore, patients were asked to indicate whether the ankle bandage provided any improvements during the execution of different biomechanical tests regarding pain and ankle stability (“much worse” to “much better”), and stability (“no” to “strong”).

#### 2.2.2. Fine Coordination and Proprioception Test

A customized foot pedal allowing inversion and eversion movements within a 5° range was used (Figure 1A). Subjects were instructed to trace a target sine wave displayed on a screen (velocity of approx. 7 cm/s) by manipulating the pedal. The extent of the target sine wave ranged from 3° inversion to 3° eversion (Figure 1B). To ensure the highest possible test quality, the rotation center of the ankle joint was aligned vertically with the pivot point of the pedal. Furthermore, to guarantee that the ankle joint, rather than the hip joint, performed the movement, it was essential that the lateral side of the lower leg remained in contact with the device throughout the test (Figure 1A). Five trials of 18 s each were performed, with the mean deviation from the target sine wave calculated for the last two trials within the 6–16 s interval.

#### 2.2.3. Single Leg Stance (Quasi-Static Postural Stability)

Subjects’ quasi-static postural stability was quantified while performing single leg stances using a pressure distribution platform (Zebris FDM 1.5; Isny, Germany, sampling frequency 100 Hz). Subjects stood on a pressure distribution platform, maintaining an upright posture with their knee straight but not locked, arms crossed in front of the chest, and gaze directed ahead. Each trial lasted 20 s, with center of pressure (COP) parameters COP length and 95% COP confidence area were measured for analysis. Longer COP excursions and larger COP areas indicate greater instability.

#### 2.2.4. Y-Balance Test (Dynamic Postural Stability)

The Y-Balance test, also known as the modified Star Excursion Balance Test (mSEBT) was used to assess subjects’ dynamic postural stability [14,43,44,45,46,47,48]. The test quantifies the ability to maintain a stable base of support, while reaching as far as possible with the lower limbs. Subjects performed the Y-Balance test standing on one foot and reaching with the other in three directions (anterior, posteromedial, posterolateral). Maximum reach distances were normalized to leg length, and any loss of balance or stance foot movement necessitated a trial repeat. All subjects completed three main trials for each direction for the three testing conditions healthy stance leg, injured stance leg with and without bandage.

#### 2.2.5. Gait

Subjects walked ten times over the pressure distribution platform integrated into a walkway. Sampling frequency was set at 100 Hz to collect sufficient data. The gait speed had to be both rapid and comfortable and could be selected by the individual. Spatial-temporal gait parameters, including gait velocity, step length, and vertical ground reaction forces (Figure 2), were recorded and calculated using the software Zebris WinFDM (version 0.1.11, Isny, Germany).

### 2.3. Statistical Analysis

Means and standard deviations (mean ± SD) were calculated for all biomechanical variables. The Shapiro-Wilk test was used to assess normal distribution. A one-way ANOVA for repeated measures followed by Bonferroni post-hoc tests was conducted to evaluate differences between conditions (healthy leg, injured leg with and without bandage). Statistical significance was set at α = 0.05. Effect sizes were calculated using Cohen’s d, categorized as trivial (<0.2), small (<0.5), medium (<0.8), or large (≥0.8).

## 3. Results

### 3.1. Demographic and Clinical Data

Table 1 presents the demographic and clinical characteristics of the subjects. The cohort included 70 subjects with a slight predominance of males (n = 36) over females (n = 34). Most injuries were to the right leg (n = 38), with the remainder affecting the left leg (n = 32). The anterior talofibular ligament was the most frequently affected structure, followed by the calcaneofibular and posterior talofibular ligaments. Commonly, injuries were associated with swelling and other general complaints.

Table 2 summarizes self-rated pain perceptions using a visual analogue scale (0–10). Subjects reported mild to moderate pain, which increased during walking compared to standing. The bandage generally alleviated pain. No significant differences in pain perception were noted between walking barefoot and in shoes.

As shown in Figure 3, subjects reported various improvements due to the bandage, with strong stabilizing effects noted by half the cohort and moderate effects by the other half. Pain improvement was rated mostly “better” (“slightly better” and “much better”). Additionally, subjects’ ratings for the motor performance tests were primarily “better”.

### 3.2. Fine Coordination and Proprioception Test

As shown in Table 3, no statistically significant differences were observed in the fine coordination and proprioception test between conditions (*p* = 0.078), although there was a trend towards smaller deviations with the ankle bandage.

### 3.3. Motor Performance

#### 3.3.1. Single Leg Stance (Quasi-Static Postural Stability)

As presented in Table 4, significant differences were found in COP length during single leg stance (*p* < 0.001). Subjects showed especially longer COP excursions for the injured leg without bandage compared to the injured leg when wearing the bandage and the healthy leg. Nevertheless, the COP 95% confidence area showed the same result as the COP length, however as a trend and not statistically significant.

#### 3.3.2. Y-Balance Test (Dynamic Postural Stability)

The Y-Balance test revealed higher values for the maximum reach distance, when subjects were standing on their healthy leg compared to when subjects were standing on their injured leg with or without bandage. These statistically significant differences with medium effect sizes were exclusively found for the anterior direction (Figure 4).

#### 3.3.3. Gait

As shown in Table 5, various statistically significant differences between the three walking conditions could be found. With respect to the healthy leg, the injured leg showed a shorter single stance phase and decreased second peak of the vertical force, regardless of wearing the bandage or not. The step length and the first peak of the vertical force did only reveal differences between the healthy and the injured leg without wearing the bandage. The step length for the healthy leg was decreased and the first peak of the vertical force was significantly higher.

When comparing walking with and without bandage, the healthy leg showed longer steps with higher vertical forces of the second peak when walking with bandage. The injured leg showed a longer relative single stance phase and higher vertical forces of the first and second peak, when walking with a bandage.

The parameters gait velocity and relative double stance phase, however, did not show any significant differences between the conditions walking with bandage and walking without bandage.

## 4. Discussion

This study aimed to investigate the effects of wearing an ankle bandage on fine coordination and proprioception, as well as on motor performance in subjects five weeks post-orthotic treatment for ankle sprains. We hypothesized that the bandage would reduce pain, improve fine coordination and proprioception, and enhance motor performance, including static and dynamic balance, as well as gait.

### 4.1. Fine Coordination and Proprioception Test

#### 4.1.1. Effects of the Sprain

No statistically significant differences were found between testing conditions for the fine coordination and proprioception test. However, we observed a trend towards worse fine coordination for the healthy leg and the best fine coordination for the injured leg when wearing the bandage. Contrary to expectations, our results showed no significant proprioceptive impairments, unlike previous studies [3,14,15,16,17,18,19,20,28,50]. Therefore, we raised the question of what could have led to better performance of the injured ankle compared to the healthy ankle. Since those studies primarily investigated proprioception in CAI rather than in the acute or subacute phase, our results might possibly be attributed to increased awareness and caution when moving the injured leg. This notion is supported by the fact that our subjects still reported pain during the test [28,51,52]. Various compensation mechanisms could may also have played an important role. Since our patients were tested in the subacute phase following the sprain, the influence of active healing processes, such as increased blood flow and tissue supply, may have helped compensate possible injury-related proprioceptive impairments, resulting in slightly better performance [31,53]. Different anatomical structures of the ankle joint also have various types of mechanoreceptors that gather afferent information to control movements. These receptors are situated within the ligaments, tendons, the joint capsule, surrounding muscles, and even the skin. Due to this redundancy, potentially unaffected structures might have compensated and reweighted the impaired function of those structures affected by the sprain [54,55,56]. Our test required subjects to move a pedal with inversion and eversion movements of the ankle joint to follow a presented target sine wave on the screen. Therefore, the results of this test might also have been dominated and possibly overcompensated by the visual system [22,54].

#### 4.1.2. Effects of the Bandage

A more plausible result of our study suggests that the injured leg with the bandage had a tendency towards better proprioceptive performance compared to without the bandage. Although the literature yields inconsistent findings regarding the effects of bandages and elastic tapes on ankle joint fine coordination and proprioception, there may be several explanations for our results [26,57,58,59,60,61]. Since the bandage mechanically stabilizes the ankle joint, it increases resistance during joint movement. Consequently, it may have assisted and restricted the inversion and eversion movements of the ankle joint, especially at the reversal points of the pedal, which led to smaller error values during our test [32,33,34,62]. This is supported by the results of the subjective data, in which the subjects ranked the effects of wearing the bandage. The vast majority of the subjects reported strong (50%) to moderate (48.6%) stabilizing effects on the ankle joint. In terms of the fine coordination and proprioception test, 15.7% of the subjects reported significant improvements, while 41.4% noted slight improvements while wearing the bandage. Another reason for enhanced proprioceptive performance using ankle supports, like bandages might have to do with the compression of the underlying musculoskeletal structures. Consequently, this might stimulate a greater number of cutaneous mechanoreceptors, as well as receptors in the joint capsule and ligaments, thereby enhancing proprioceptive acuity and ankle movements [3,26,51,57,58]. Given that we only found a trend towards better proprioceptive performance, and that only few studies have investigated the effects of ankle support on proprioception in the acute or subacute phase rather than in CAI, the benefits of bandages on fine coordination and proprioception still appear elusive. In addition, our testing setup differed from that of other studies, which used detection of passive motion or active joint position tests. Therefore, comparing our results to those of other studies should be done carefully.

### 4.2. Motor Performance

#### 4.2.1. Effects of the Sprain

In terms of motor performance, the injured leg showed significant impairments compared to the healthy leg. This was demonstrated by higher sway values for the single leg stances, reduced reach distance for the Y-Balance test, and impaired gait performance with longer steps, shorter single stance phases and smaller vertical ground reaction forces.

Two main causes might help to explain those findings: mechanical ankle instability (MAI) and functional ankle instability (FAI). Although those terms have primarily been used to describe the development of CAI, they also apply to the acute and subacute phase of ankle injuries, which was analyzed in our study. While MAI is associated with complaints of mechanical instability and laxity, as well as pain and swelling, FAI is more related to impaired functional muscle control due to compromised proprioceptive and sensory structures [28,47,63,64]. Although both causes interact and overlap each other, higher sway values during quasi-static balance tasks, such as single leg stances, might be explained predominantly by MAI [28,53,65]. That makes sense, considering that our patients reported complaints of laxity and instability. Our results are in line with various other studies [19,28,66,67,68]. For example, Hertel et al. reported that during single leg stances postural sway length and velocity increased significantly on the injured leg as compared with the uninjured leg [66]. Pourkazemi et al., even stated that single leg stances most strongly discriminated between participants with ankle sprains and healthy control subjects [67]. These results may be due to changes in postural strategies. Maintaining balance is usually accomplished by using either the ankle strategy or the hip strategy. The ankle strategy shifts the center of gravity by moving the entire body as a single-segmented inverted pendulum around the ankle joint, whereas the hip strategy involves moving the body as a double-segmented inverted pendulum with counter-phase motions around both the hip and ankle joints [24,66,67,69]. As a result of the injury, the subjects may switch from the typical ankle strategy to the hip strategy, which is less effective for quasi-static balance tasks [67,70,71]. Nevertheless, there are also several studies, which contradict our study findings for the single leg stance test [31,72,73]. Possible explanations for this contradiction might include varying methodological factors, such as differences in test paradigms and study groups. Most of those studies investigated patients with CAI, however not patients in the acute or subacute phase of the sprain. As suggested by Ross and Guskiewicz, quasi-static balance tests might have limited sensitivity, because they only assesses a single component of balance and therefore should be supplemented by additional, more challenging measures, like the Y-Balance test [67,73].

The Y-Balance test quantifies the ability to maintain a stable base of support while reaching as far as possible with the lower limbs. In our study, the maximum reach distance was significantly reduced when subjects were standing on their injured leg without a bandage compared to when they were standing on the healthy leg. However, we only found this for the anterior direction, but not for the posterior-medial or posterior-lateral directions. While the literature yields inconsistent results regarding the most impaired reaching direction, some studies have also reported impairments in the anterior direction [45,48,67,74]. For example, Pourkazemi et al. reported that a reduced anterior reach distance most strongly discriminated between subjects with ankle sprains and healthy subjects [67]. Similar results have also been found in the study by McCann et al., in which the study group with CAI achieved lower anterior test scores compared to subjects coping with lateral ankle sprains [74]. Since the Y-Balance test is considered a measure of dynamic postural stability, patients’ diminished reach distance might predominantly be related to FAI [67]. This is supported by other studies that have reported altered lower limb muscle activity in patients with CAI when performing anterior reaches [75,76]. Both studies found less activity for the tibialis anterior and peroneus longus muscles compared to patients coping with CAI. Another key factor contributing to patients’ impaired anterior reach distance could be altered kinematics resulting from reduced ROM of the ankle joint [43,44,67,77]. In the studies from Pourkazemi et al. and Basnett et al., subjects with CAI exhibited restricted dorsiflexion motion in comparison to healthy control groups. Patients’ limited ROM moderately correlated with the anterior reach distance of the Y-Balance test [67,77]. It has also been reported that individuals with CAI showed reduced hip and knee flexion while executing the test in the anterior direction. Furthermore, knee flexion and torso rotation have been identified as the primary kinematic predictors of reach distance during performance in the anterior direction [44,74]. Hence, limited ankle dorsiflexion ROM is a strongly limiting factor for anterior performance in the Y-Balance test [43,47,78]. However, other authors have reported significantly diminished reach distances not only for the anterior direction, but also for the posterior-lateral [43,45,46,48,76] and posterior-medial [46,79] directions among patients with CAI when compared to healthy subjects, CAI copers, or the injured and uninjured sides. Similar to the anterior direction, diminished reach distance in both posterior directions might include reduced activity of lower limb muscles [75,76,80], as well as altered kinematics due to restricted dorsiflexion ROM [43,44,45,46,74]. Other factors may include pain and fear of falling due to ankle instability. Although the results are controversial, proprioceptive and neuromuscular deficits might also play a role [28,43,45,51,55,57,66].

During walking, we found significant differences in spatial-temporal parameters for the injured leg compared to the healthy leg. More specifically, the injured leg showed longer steps and shorter single stance phases during walking. Since longer steps typically signify higher gait quality, our observation appears counter intuitive at first glance. It even seems to contradict various other studies in which reduced step lengths were reported for patients with ankle injuries [52,81,82,83]. However, there is a simple methodological explanation for this contradiction. The mentioned studies conducted inter-individual investigations, comparing a group with ankle injuries to another group of healthy subjects. In contrast, we observed intra-individual effects of the ankle injury by comparing the injured leg with the healthy leg within the same subjects. As the healthy leg compensates for impairments of the injured leg, patients in our study attempted to extend the step length of their injured leg to increase swing time and consequently minimize the loading time of the injured leg. This is supported by our findings of reduced single stance phases for the injured leg during walking [52,81,82,83]. Since our patients were tested in the subacute phase of the sprain, this effect was likely driven by the pain they reported, as well as by fear of falling [52,82]. Other spatial-temporal parameters reported in the literature to characterize impaired gait performance in patients with ankle sprains include reduced walking speed, decreased cadence, and wider steps [52,81,82,83,84]. Other studies also reported a decrease in vertical foot-floor clearance before heel strike, an increased inversion velocity during heel strike, reduced maximum plantar flexion during the stance phase, and a more inverted foot position throughout the entire gait cycle [43,52,63,84,85,86]. The altered kinematics of the patients may help to explain the differences we found for the kinetic parameters comparing the injured and healthy legs. We observed diminished ground reaction forces for the injured leg for the first peak, corresponding to the loading response, and the second peak, corresponding to the terminal stance phase of the gait cycle [49]. Accordingly, our results are consistent with those of several other studies [52,87,88]. For example, Nyska et al. reported reduced impact at the beginning and end of the stance phase with a significant reduction in the relative forces under the heel and toes in subjects with CAI during walking. The authors also reported slower weight transfer from the heel to toe, and a lateral shift of the foot’s COP, possibly caused by a more inverted foot position [88]. Similarly, in the studies by Punt et al. and Doherty et al., the authors observed decreased maximum power and reduced maximum moments in patients with ankle sprains compared to healthy individuals [52,87]. The findings of patients unloading their injured leg during walking contrast those of Koldenhoven et al., who found increased ankle plantarflexion moments during the late stance phase to toe-off [84]. Nevertheless, this may be due to differences in study methods. In their study, subjects walked on a split belt treadmill and wore standardized shoes, whereas patients in our study, patients were barefoot and walked across a pressure distribution platform. Furthermore, they tested patients with CAI, whereas our patients were tested in the subacute phase of injury and reported pain during walking. The impaired gait pattern from patients with ankle sprains is considered to have a multifactorial pathology and can be attributed to several co-existing factors. This includes mechanical instability, proprioceptive impairments, neuromuscular control deficits, postural instability, reduced ROM of the ankle joint, altered activation of lower limb muscles, as well as pain and fear [17,19,28,43,52,53,55,63,67,81,82,84,85,86,89,90,91,92].

#### 4.2.2. Effects of the Bandage

Wearing the bandage in the subacute phase of the sprain significantly enhanced our subjects’ single leg stance performance and normalized gait. However, the bandage did not improve our subjects’ reach distance for the Y-Balance test.

Our findings of reduced sway when wearing the bandage during single leg stances are supported by two studies from Hadadi et al. In those studies, the authors concluded that both the immediate use of soft or semi-rigid ankle braces and their continuous use for four weeks improved single leg stance performance in subjects with CAI [14,93]. Small but significant and effective benefits on single leg stance in patients with ankle injuries using soft or semi-rigid orthosis have also been found in the studies by Best et al. and Faraji et al. [3,31]. Moreover, in athletes with functional ankle instability, Baier and Hopf found that a flexible ankle orthosis significantly reduced the medio-lateral sway velocity during single leg stances and changed the sway pattern by reducing the percentage of linear movements [94]. The positive effects of ankle support on quasi-static balance tasks, such as single leg stance, may predominantly be explained by proprioceptive stimulation and mechanical support. By stimulating cutaneous mechanoreceptors and exerting pressure on underlying musculoskeletal structures, ankle supports might offer additional sensory input about joint position and movements. Therefore, ankle supports may help detect internal balance perturbations, and thereby improve control of postural sway in individuals with ankle sprains [3,14,94,95]. The observation that wearing ankle supports decreases postural sway in patients more than in healthy subjects reinforces this theory [94,95,96]. Apart from proprioception, a noticeable portion of the improvements could be attributed to mechanical stabilization [14,51,94,97]. The study by Thonnard et al., who investigated the inversion torque of bare and braced ankles under static and dynamic conditions using a customized mechanical apparatus, supports this finding. In their study, they found that the additional inversion ankle torque generated by an elastic brace effectively increased the passive resistance against ankle inversion movement compared with the bare ankle tests. Although the additional torque provided by the braces was small relative to the torques and forces applied to the foot during a typical sprain situation, it might contribute to additional stability during single leg stance, resulting in reduced sway [33,62]. That also aligns with findings that subjects with ankle injuries report feeling more stable and comfortable during balance tests when wearing orthotics [95,97]. In addition, the vast majority of subjects in our study reported strong (50%) to moderate (48.6%) stabilizing effects on the ankle joint when wearing the bandage. For the single leg stance test, 28.6% of the subjects reported strong improvements, while 57.1% reported slight improvements when wearing the bandage. Furthermore, reduced pain perception when wearing the bandage might have had a positive effect on postural sway.

In our assessment of dynamic postural stability using the Y-Balance test, we did not observe any improvements in patients’ reach distances while wearing the bandage, regardless of the testing direction. This was surprising, since wearing the bandage enhanced single leg stance performance and gait in our study. Alawna et al. investigated the effects of ankle taping and bandages on the reach distance of one-hundred patients with CAI. They conducted measurements at baseline, immediately after support, and then at 2 weeks and 2 months post-support. Their results showed that ankle taping and bandaging does not immediately improve reach distance, but that improvements occurred after 2 weeks and 2 months [57]. Moreover, Hadidi et al. investigated the effects of ankle taping and soft or semi-rigid ankle braces on the reach distance of the Y-Balance test before and after a 4-week intervention period. Their findings showed that the use of tape and a soft or a semi-rigid ankle brace for 4 weeks were all beneficial in improving the reach distance in individuals with CAI [14]. A study by John et al. investigated the effects of an elastic ankle support on dynamic balance in patients with CAI using the Y-Balance Test. The authors concluded that the acute use of elastic ankle support was ineffective in enhancing dynamic balance [98]. Considering all of the presented results, it seems that patients may need some time to adapt to the bandage in order to fully experience the positive effects on mechanically stabilizing the ankle joint and enhancing proprioception for this specific test. This is conceivable, as the Y-balance test is particularly challenging and differs from more daily activities, such as standing on one leg or walking. Since we tested patients in the subacute phase, their pain and fear of falling might have been more severe compared to studies involving patients with CAI, potentially limiting the effectiveness of the bandage condition in our study.

Regarding gait, we were surprised to find only one study investigating the influence of orthotic support in patients with ankle injuries [83]. In this study, 10 subjects with CAI walked without a brace, with a flexible brace, and with a semi-rigid ankle brace while their kinematics and kinetics were recorded using a marker-based system and a force plate. Although the effects were small, few differences were measured between the brace and no brace conditions. In summary, the authors described the effect of wearing braces during walking, noting altered foot angles at the heel strike and toe-off, altered braking forces, reduced step lengths, and a reduction in the stance phase. Therefore, this study partly confirms our findings, of extended stance phases and higher vertical ground reaction forces for the loading response and the terminal stance phase when walking with the bandage compared to walking without the bandage. Noteworthy, wearing the bandage during walking also improved our subjects’ healthy leg performance, resulting in longer steps and higher vertical ground reaction forces during the terminal stance phase. Consequently, wearing the bandage might help reduce asymmetry, which could potentially mitigate the risk of injuries [99,100]. The proposed mechanisms explaining the effectiveness of ankle orthoses during walking include mechanical support, improving proprioceptive and sensorimotor function, as well as enhancing ankle positioning and muscular efficiency around the ankle joint [14,51,62,97,101]. Spaulding et al., suggested that ankle braces affect forward progression without significantly impacting gait characteristics or causing compensatory or adaptive motion elsewhere in the lower limb [83]. Specifically, the reduction of pain when wearing the bandage might have encouraged our patients to exert more and longer-lasting loads on the injured leg during the single stance phase of the gait.

When interpreting our results, some limitations should be taken into account. First, our results can only be compared to other studies that used a similar study design. Since a wide variety of ankle supports has been investigated in the literature, studies that utilize soft ankle supports may be most comparable to the ankle bandages used in our study. Therefore, it should also be noted that we focused on patients in the subacute phase of ankle sprains, while most other studies investigated the effects of ankle support in patients suffering from CAI. Specific limitations of this study include the absence of MRI data to precisely identify injury types and the lack of kinematic and electromyographic analyses. Future research should explore these aspects and compare the efficacy of different types of ankle supports.

## 5. Conclusions

Five weeks post-ankle sprain, subjects exhibited mild to moderate pain and ankle instability, with impaired motor performance in the injured leg compared to the healthy leg. This was evident from higher postural sway during single leg stances, reduced reach distances in the Y-Balance test, and altered gait characterized by longer steps, shorter stance phases, and reduced vertical ground reaction forces. Contrary to expectations, no significant impairments were observed in the fine coordination and proprioception tests for the injured leg. Subjects reported moderate to strong improvements in ankle stability and pain relief while wearing the bandage. The bandage significantly improved single leg stance performance, normalized single stance phase duration, and increased vertical ground reaction force during walking. The impact of the bandage on fine coordination and proprioception is not clearly stated. In conclusion, wearing a bandage directly after the acute phase of an ankle sprain significantly enhances motor performance, particularly in standing and walking tasks, despite having no clear effect on fine coordination and proprioception. These findings suggest that ankle bandages can be a valuable adjunct in the early rehabilitation phase to improve motor performance and reduce pain, potentially preventing further complications and enhancing recovery.

## Figures and Tables

**Figure 1 life-14-00810-f001:**
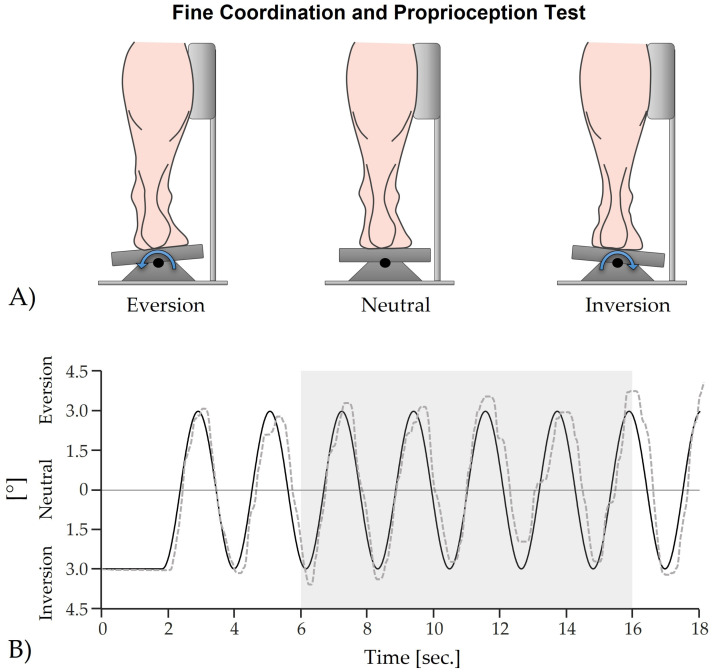
Fine coordination and proprioception test; (**A**) Hardware: Foot pedal for inversion and eversion movements of the ankle joint; (**B**) Software: Target sine wave (solid black line) which gets traced by a random subject, which was wearing the bandage (dashed grey line). The grey box indicates the analyzed area, in which the mean deviation between both signals was calculated.

**Figure 2 life-14-00810-f002:**
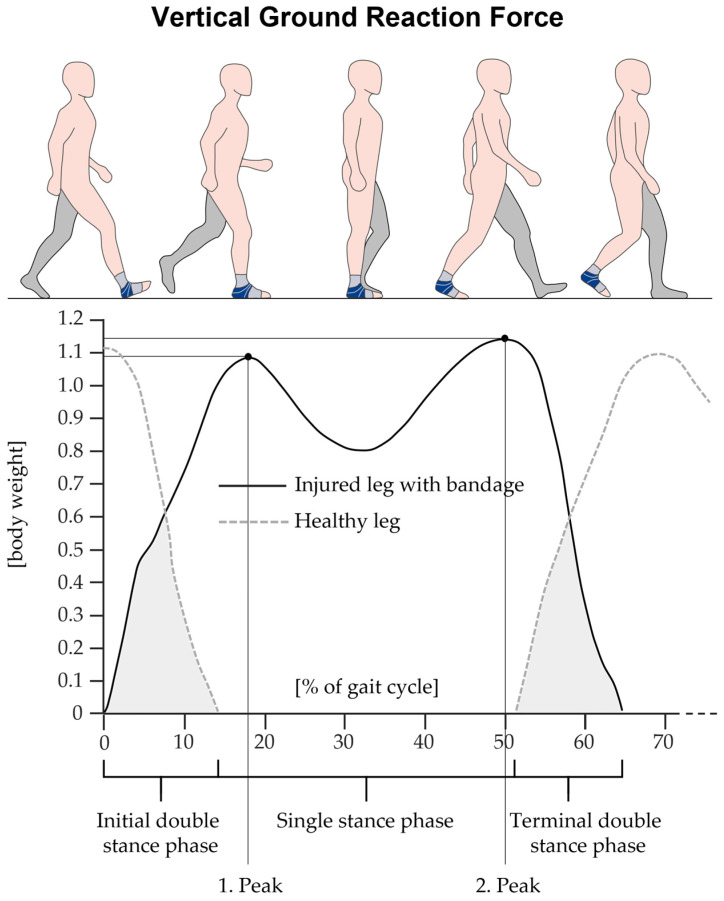
Illustration of an actual vertical ground reaction force curve during walking from a random subject. The black solid line indicates the curve of the ground reaction force for the injured leg with a bandage, while the grey dashed line indicates the curve for the healthy leg. Shown are the first peak, corresponding to the loading response, and the second peak, corresponding to the terminal stance phase of the gait cycle [49]. The illustration also displays the relative duration of the single stance phase and both double stance phases (grey area). Note that for the statistical analysis, the relative double stance phase is the sum of both the initial and terminal double stance phases.

**Figure 3 life-14-00810-f003:**
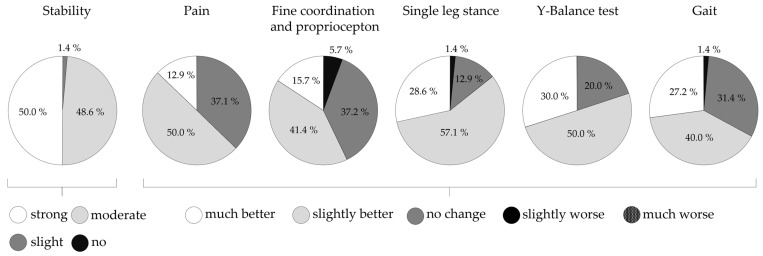
Self-ranked improvements due to wearing the bandage, data presented as percentage of n.

**Figure 4 life-14-00810-f004:**
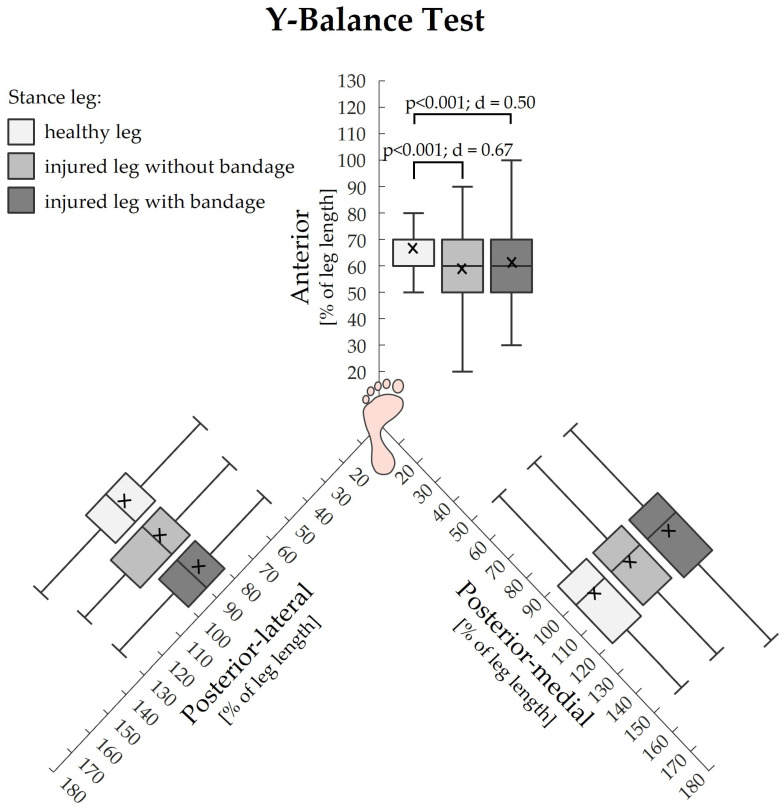
Comparisons of the three Y-Balance test directions between all three conditions healthy stance leg, injured stance leg without and with bandage. Data presented as percentage of the leg length. The cross additionally marks the mean value.

**Table 1 life-14-00810-t001:** Demographic and clinical data (structures of the ankle joint with pressure pain), presented as mean ± SD.

Age [Years]	Height [cm]	Weight [kg]	Gender	Side of the Injured Leg	Pain Rating	Instability Rating
34.8 ± 11.8	173.3 ± 10.1	78.7 ± 16.4	male 36; female 34	left 32; right 38	1.6 ± 1.3	3.0 ± 2.2
**Injured structures of the ankle joint**			**n**	**Relative n to total number of subjects (70) [%]**		
Anterior talofibular ligament			54	84.4		
Posterior talofibular ligament			31	48.4		
Calcaneofibular ligament			38	59.4		
Swelling			53	82.8		
General complaints			61	95.3		

**Table 2 life-14-00810-t002:** Self-rated pain perception of the injured ankle joint (visual analogue scale from 0 to 10). Data are presented as mean ± SD, minimum and maximum.

Pain	Walking Barefoot	Standing Barefoot	Walking in Shoes	Standing in Shoes	Walking with Bandage	Standing with Bandage
mean ± SD	1.60 ± 1.58	1.13 ± 1.45	1.69 ± 1.45	1.09 ± 1.27	1.18 ± 1.46	0.88 ± 1.25
range [Min Max]	[0 7]	[0 7]	[0 6]	[0 5]	[0 6]	[0 6]

**Table 3 life-14-00810-t003:** Comparison of the deviation value between the target sine wave and the signal drawn by the subjects between all three conditions healthy leg, injured leg with and without bandage. Data presented as mean ± SD.

Parameter	Healthy Leg	Injured Leg without Bandage	Injured Leg with Bandage	*p*-Value	d
Mean deviation [°]	1.30 ± 0.5	1.26 ± 0.5	1.20 ± 0.5	0.078	-

**Table 4 life-14-00810-t004:** Comparison of the COP parameters between all three conditions healthy leg, injured leg with and without ankle bandage. Data presented as mean ± SD. Statistically significant differences are indicated with a, b.

Parameter	Healthy Leg	Injured Leg without Ankle Bandage	Injured Leg with Ankle Bandage	*p*-Value	d
COP length [mm]	470.0 ± 115.9 a	548.8 ± 145.1 a;b	477.5 ± 106.7 b	<0.001 a < 0.001 b < 0.001	1.1 a 0.60 b 0.56
COP 95% confidence area [mm^2^]	217.6 ± 101.5	238.2 ± 91.1	216.3 ± 79.0	0.263	-

**Table 5 life-14-00810-t005:** Intra-group and inter-group comparisons (between the healthy leg and the injured leg for the conditions walking with and without bandage) of the spatial-temporal and kinetic gait parameters. Data are presented as mean ± SD. Statistically significant differences are indicated with a, b, c, d.

Parameter	Walking Injured Leg without Bandage		Walking Injured Leg with Bandage		*p*-Value	d
gait velocity [km/h]	3.98 ± 0.67		4.01 ± 0.59		0.251	-
rel. double stance phase [% of gait cycle]	29.17 ± 4.20		29.46 ± 4.22		0.121	-
Parameter	healthy leg	Injured leg without bandage	healthy leg	Injured leg with bandage	*p*-value	d
rel. single stance phase [% of gait cycle]	65.3 ± 2.7 a	64.1 ± 2.5 a;b	65.4 ± 3.2 c	64.7 ± 2.9 b;c	a < 0.001 b 0.018 c 0.013	a 0.46 b 0.22 c 0.24
step length [cm]	57.7 ± 6.1 a;b	61.1 ± 7.8 a	61.1 ± 8.5 b	61.0 ± 7.3	a < 0.001 b < 0.001	a 0.49 b 0.46
1. Peak of the vertical force [body weight]	1.07 ± 0.07 a	1.05 ± 0.06 a;b	1.07 ± 0.07	1.07 ± 0.07 b	a 0.004 b 0.004	a 0.31 b 0.31
2. Peak of the vertical force [body weight]	1.11 ± 0.07 a;b	1.08 ± 0.07 b;c	1.13 ± 0.06 a;d	1.11 ± 0.06 c;d	a 0.039 b < 0.001 c < 0.001 d 0.001	a 0.31 b 0.43 c 0.33 d 0.31

## Data Availability

The dataset used and analyzed in this study is available from the corresponding author upon reasonable request.

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
