# Peer review of "Immediate Effects of Wearing an Ankle Bandage on Fine Coordination, Proprioception, Balance and Gait in the Subacute Phase of Ankle Sprains"

_life, 2024, doi:10.3390/life14070810_

Round 1

Reviewer 1 Report

Comments and Suggestions for Authors

For Title and Abstract:

1.         Clarity and Focus:

Problem: The introduction lacks a clear definition of the research problem and the hypothesis being tested.

Improvement: Clearly state the research question and the hypothesis in the abstract. For example: "This study investigates the hypothesis that wearing an ankle bandage immediately after an acute ankle injury improves fine coordination, proprioception, and motor performance."

2.         Specificity and Detail:

Problem: The abstract is vague about the specifics of the tests conducted and the results obtained.

Improvement: Provide more details on the methodology and results. For example: "Biomechanical investigations included fine coordination and proprioception tests, single leg stances, the Y-Balance test, and gait analysis. Results indicated moderate to strong improvements in ankle stability and pain relief while wearing the bandage."

3.         Consistency:

Problem: The text inconsistently references the timing of the intervention.

Improvement: Ensure consistent terminology regarding the timing. For example: "Subjects were tested five weeks post-injury to assess the immediate effects of the ankle bandage."

4.         Impact of Findings:

Problem: The significance of the findings is not clearly articulated.

Improvement: Emphasize the implications of the findings for clinical practice. For example: "These findings suggest that ankle bandages may play a crucial role in early-stage rehabilitation by enhancing motor performance and reducing pain."

5.         Comparative Analysis:

Problem: There is no comparative analysis with existing literature.

Improvement: Compare your findings with previous studies. For example: "Unlike previous studies that focused on long-term effects, our study highlights the immediate benefits of ankle bandages in the acute phase of injury."

6.         Language Precision:

Problem: Some sentences are unclear or awkwardly phrased.

Improvement: Use precise language and revise awkward sentences. For example, revise "All subjects completed several rating questionnaires and biomechanical investigations" to "Subjects completed rating questionnaires and underwent comprehensive biomechanical assessments."

7.         Statistical Significance:

Problem: The abstract does not mention the statistical significance of the results.

Improvement: Include information about the statistical significance of the findings. For example: "Wearing the bandage significantly normalized single leg stance performance (p<0.05), stance phase duration (p<0.05), and vertical ground reaction forces during walking (p<0.05)."

8.         Scope and Limitations:

Problem: The abstract does not address the limitations of the study.

Improvement: Briefly mention the limitations. For example: "Further research is needed to explore the long-term effects and to confirm these findings in larger, more diverse populations.

For Introduction

Clarity and Focus:

Problem: The introduction lacks a clear, concise statement of the research problem and objective.

Improvement: Explicitly state the research problem and objective at the beginning. For example: "This study addresses the gap in research on the immediate effects of ankle bandages during the acute phase of ankle sprains, specifically focusing on their impact on fine coordination, proprioception, and motor performance."

Logical Flow:

Problem: The introduction has a disjointed flow, making it difficult to follow the progression of ideas.

Improvement: Organize the information logically, ensuring a clear transition between sections. For example, start with the anatomy and function of the ankle, move to the types and consequences of ankle sprains, and then discuss the treatment and the current research gap.

Specificity and Detail:

Problem: Some parts of the introduction are vague or overly broad.

Improvement: Provide specific details where necessary. For example: "Ankle sprains mostly occur following uncontrolled, sudden movements, such as plantarflexion and inversion, leading to excessive force or strain on the lateral ligaments of the joint."

Redundancy:

Problem: The introduction contains redundant information.

Improvement: Eliminate redundant sentences. For example, merge the sentences discussing the classification of ankle sprains by severity and duration to avoid repetition.

Comparative Analysis:

Problem: There is a lack of comparative analysis with previous studies.

Improvement: Compare and contrast your study with previous research more explicitly. For example: "Unlike previous studies that have focused on the effects of bandages in healthy subjects or those with chronic ankle instability, our study examines their immediate impact during the acute phase of injury."

Language Precision:

Problem: Some sentences are unclear or awkwardly phrased.

Improvement: Use precise and clear language. For example, revise "During running and jumping, those forces can reach multiple times the body weight" to "During running and jumping, the forces exerted on the ankle joint can exceed several times the individual's body weight."

Significance of the Study:

Problem: The significance of the study is not clearly articulated.

Improvement: Clearly state the significance of your research. For example: "Understanding the immediate effects of ankle bandages can inform early intervention strategies, potentially reducing the incidence of chronic ankle instability and improving patient outcomes."

Statistical Data:

Problem: The introduction provides statistics without sufficient context or explanation.

Improvement: Integrate statistical data with explanatory context. For example: "Ankle sprains account for approximately 49% of all injuries in recreational athletes, highlighting the need for effective early treatment strategies."

Hypothesis:

Problem: The hypothesis is mentioned but not clearly integrated into the introduction.

Improvement: Clearly state the hypothesis in the context of the research gap. For example: "We hypothesize that wearing an ankle bandage immediately after an acute lateral ankle sprain will reduce pain, enhance fine coordination and proprioception, and improve motor performance, including balance and gait."

For Materials and Methods

1.         Participant Description and Inclusion Criteria:

Problem: The inclusion and exclusion criteria are not clearly and comprehensively stated.

Improvement: Clearly define the inclusion and exclusion criteria with more detail and structure. For example: "Inclusion criteria were adults aged 18-60 years presenting with an acute unilateral supination ankle sprain (minimum grade 2) within three days post-injury. Exclusion criteria included grade 1 ankle sprains, upper leg sprains within the last 12 months, acute concomitant osseous injuries, history of confirmed lateral ligament injury, chronic ankle instability (CAI), neurological dysfunctions, rheumatic diseases, gout, arthrosis, recent surgeries, use of anticoagulants or corticosteroids, and any other conditions affecting motor performance or proprioception."

2.         Experimental Setup and Randomization:

Problem: The description of the experimental setup lacks detail on randomization and control measures.

Improvement: Specify how randomization was implemented and how control measures were maintained. For example: "All biomechanical tests were conducted on both the injured and healthy legs in a randomized order to minimize the influence of fatigue and habituation. Subjects performed familiarization trials with the bandage and tasks before the main trials."

3.         Detailed Procedures for Biomechanical Tests:

Problem: The procedures for the biomechanical tests are not described with sufficient clarity and precision.

Improvement: Provide detailed and precise descriptions of each test. For example:

Fine Coordination and Proprioception Test: "A customized foot pedal allowing inversion and eversion movements within a 5° range was used. Subjects were instructed to trace a target sine wave displayed on a screen by manipulating the pedal. Five trials of 18 seconds each were performed, with the mean deviation from the target sine wave calculated for the last two trials within the 6-16 second interval."

Single Leg Stance: "Subjects stood on a pressure distribution platform, maintaining an upright posture with their knee straight but not locked, arms crossed in front of the chest, and gaze directed ahead. Each trial lasted 20 seconds, with center of pressure (COP) parameters measured for analysis."

Y-Balance Test: "Subjects performed the Y-Balance test standing on one foot and reaching with the other in three directions (anterior, posteromedial, posterolateral). Maximum reach distances were normalized to leg length, and any loss of balance or stance foot movement necessitated a trial repeat."

Gait Analysis: "Subjects walked ten times over a pressure distribution platform integrated into a walkway. Spatial-temporal gait parameters, including gait velocity, step length, and vertical ground reaction forces, were recorded."

4.         Subjective Ratings:

Problem: The method for collecting subjective ratings is not clearly described.

Improvement: Clearly describe the process and tools used for subjective ratings. For example: "Subjects provided anthropometric data and rated their pain and stability improvements while walking and standing barefoot, in shoes, and with the bandage using visual analogue scales (0-10)."

5.         Statistical Analysis:

Problem: The statistical analysis section is brief and lacks detail.

Improvement: Expand the statistical analysis section to include more details on the methods used. For example: "Mean and standard deviations were calculated for all biomechanical variables. The Shapiro-Wilk test was used to assess normality. A one-way ANOVA for repeated measures followed by Bonferroni post-hoc tests was conducted to evaluate differences between conditions (healthy leg, injured leg with and without bandage). Statistical significance was set at α = 0.05. Effect sizes were calculated using Cohen's d, categorized as trivial (< 0.2), small (< 0.5), medium (< 0.8), or large (≥ 0.8)."

6.         Clarity and Precision:

Problem: Some sentences are unclear or awkwardly phrased.

Improvement: Use precise and clear language throughout. For example, revise "The bandage, which was used in this study (MalleoTrain® Bauerfeind AG), consisted of an elastic but tight-fitting high-low knit" to "The MalleoTrain® bandage (Bauerfeind AG) used in this study is made of an elastic, tight-fitting high-low knit material, providing alternating pressure massage during movement."

For Results:

1.         Demographic and Clinical Data Presentation:

Problem: The presentation of demographic and clinical data is unclear and lacks detail.

Improvement: Present demographic and clinical data in a clear and detailed manner. For example: "Table 1 presents the demographic and clinical characteristics of the subjects. The cohort included 70 subjects with a slight predominance of males (n=36) over females (n=34). Most injuries were to the right leg (n=38), with the remainder affecting the left leg (n=32). The anterior talofibular ligament was the most frequently affected structure, followed by the calcaneofibular and posterior talofibular ligaments. Commonly, injuries were associated with swelling and other general complaints."

2.         Statistical Analysis and Interpretation:

Problem: The statistical analysis is not clearly explained, and the significance of the results is not fully interpreted.

Improvement: Clearly explain the statistical methods and interpret the results. For example: "The Shapiro-Wilk test confirmed the normal distribution of variables. One-way ANOVA for repeated measures followed by Bonferroni post-hoc tests identified statistically significant differences between conditions. Effect sizes (Cohen’s d) were calculated to quantify the magnitude of differences, with d ≥ 0.8 considered large."

3.         Subjective Ratings:

Problem: The description of subjective pain ratings is vague.

Improvement: Provide more context and detail. For example: "Table 2 summarizes self-rated pain perceptions using a visual analogue scale (0-10). Subjects reported mild to moderate pain, which increased during walking compared to standing. The bandage generally alleviated pain. No significant differences in pain perception were noted between walking barefoot and in shoes."

4.         Figures and Tables:

Problem: Figures and tables are referenced without proper integration into the text.

Improvement: Integrate figures and tables into the text with clear references. For example: "As shown in Figure 3, subjects reported various improvements due to the bandage, with strong stabilizing effects noted by half the cohort and moderate effects by the other half. Pain improvement was mostly moderate, with notable improvements in single leg stance and Y-Balance tests."

5.         Results Interpretation:

Problem: The interpretation of results lacks depth and clarity.

Improvement: Provide a more detailed interpretation of the results. For example: "No statistically significant differences were observed in the fine coordination and proprioception test between conditions (p=0.078), although there was a trend towards smaller deviations with the bandage. Significant differences were found in COP length during single leg stance (p<0.001), indicating improved stability with the bandage."

6.         Data Consistency:

Problem: Inconsistent data presentation and lack of detail in some areas.

Improvement: Ensure consistency and completeness in data presentation. For example, consistently report means ± standard deviations and provide comprehensive details in tables: "Table 3 presents the mean deviation in degrees from the target sine wave across conditions. While no significant differences were detected, the trend suggests potential benefits of the bandage."

7.         Use of Technical Language:

Problem: Some technical terms and statistical measures are not well explained.

Improvement: Clearly explain technical terms and statistical measures. For example: "COP (center of pressure) parameters, including COP length and 95% confidence area, were used to assess postural stability. Longer COP excursions indicate greater instability, while smaller confidence areas suggest improved stability."

8.         Comprehensive Reporting:

Problem: The results section lacks comprehensive reporting on some tests.

Improvement: Provide a complete report on all tests conducted. For example: "The Y-Balance test revealed significantly higher reach distances on the healthy leg compared to the injured leg with or without the bandage, particularly in the anterior direction (Figure 4). Gait analysis showed no significant differences in gait velocity and double stance phase between conditions (Table 5)."

For Discussion:

1.         Objective and Hypothesis Clarity:

Problem: The aim and hypothesis are not stated clearly and concisely.

Improvement: Clearly and concisely state the aim and hypothesis. For example: "This study aimed to investigate the effects of wearing an ankle bandage on fine coordination, proprioception, and motor performance in subjects five weeks post-orthotic treatment for acute ankle injuries. We hypothesized that the bandage would reduce pain, improve fine coordination and proprioception, and enhance motor performance, including static and dynamic balance as well as gait normalization."

2.         Section Organization:

Problem: The sections are not well-organized, making it difficult to follow the results and discussion.

Improvement: Clearly separate and organize sections for better readability. For example:

           4.1. Fine Coordination and Proprioception Test

           4.1.1. Effects of the Injury

           4.1.2. Effects of the Bandage

           4.2. Motor Performance

           4.2.1. Effects of the Injury

           4.2.2. Effects of the Bandage

3.         Statistical Results Reporting:

Problem: The statistical results are not reported with enough detail or clarity.

Improvement: Clearly report the statistical results, including p-values and effect sizes where applicable. For example: "No statistically significant differences were found for the fine coordination and proprioception test between conditions (p > 0.05), although there was a trend towards better performance with the bandage (d = 0.2)."

4.         Comparison to Existing Literature:

Problem: The comparison to existing literature is vague and lacks detail.

Improvement: Provide specific comparisons to existing literature. For example: "Our findings contrast with studies on chronic ankle instability (CAI) which typically show impaired proprioceptive abilities post-injury [3,13-18,29,51,52]. This discrepancy may be due to the acute phase of injury in our study, where heightened awareness and active healing processes may play a role."

5.         Mechanisms and Theoretical Explanations:

Problem: The mechanisms and theoretical explanations are not well-developed.

Improvement: Provide detailed theoretical explanations for the findings. For example: "The trend towards better performance with the bandage may be due to increased mechanical stability and proprioceptive feedback from cutaneous mechanoreceptors [3,26,53,59,60]. The bandage may also provide compression, stimulating underlying structures and enhancing proprioceptive acuity."

6.         Subjective Data Integration:

Problem: Subjective data is mentioned but not integrated well into the discussion.

Improvement: Integrate subjective data into the discussion. For example: "Subjective reports indicated that 50% of subjects experienced strong stabilizing effects with the bandage, which aligns with the observed trends towards improved proprioceptive performance."

7.         Discussion of Motor Performance:

Problem: The discussion on motor performance is fragmented and lacks depth.

Improvement: Discuss motor performance results in depth, linking them to theoretical frameworks. For example: "Motor performance impairments in the injured leg, such as increased sway and reduced reach distance, may be attributed to mechanical ankle instability (MAI) and functional ankle instability (FAI), affecting proprioceptive and neuromuscular control [29,48,65,66]."

8.         Gait Analysis Clarity:

Problem: The gait analysis results are not clearly explained.

Improvement: Provide a clear explanation of gait analysis results. For example: "Gait analysis revealed longer steps and shorter single stance phases for the injured leg, suggesting compensatory mechanisms to minimize load on the injured leg. This aligns with findings of reduced ground reaction forces in the loading and terminal stance phases [54,83-85]."

9.         Limitations and Future Research:

Problem: Limitations and suggestions for future research are not well-articulated.

Improvement: Clearly state the limitations and future research directions. For example: "Limitations of this study include the absence of MRI data to precisely identify injury types and the lack of kinematic and electromyographic analyses. Future research should explore these aspects and compare the efficacy of different types of ankle supports."

10.       Overall Language Precision:

Problem: The language is sometimes unclear and imprecise.

Improvement: Use precise and clear language throughout. For example, revise "These results were somewhat surprising since various other studies have shown impaired proprioceptive abilities" to "Contrary to expectations, our results showed no significant proprioceptive impairments, unlike previous studies on CAI [3,13-18,29,51,52]."

Conclusion:

1.         Summary of Findings:

Problem: The summary of findings is not concise and lacks clarity.

Improvement: Clearly and concisely summarize the key findings. For example: "Five weeks post-ankle injury, subjects exhibited mild to moderate pain and ankle instability, with impaired motor performance in the injured leg compared to the healthy leg. This was evident from higher sway values during single leg stances, reduced reach distances in the Y-Balance test, and altered gait characterized by longer steps, shorter stance phases, and reduced vertical ground reaction forces."

2.         Details on Impairments:

Problem: The description of impairments lacks detail and specificity.

Improvement: Provide specific details about the impairments. For example: "Motor performance impairments included significant increases in postural sway during single leg stances and a notable reduction in reach distances in the Y-Balance test, indicating decreased dynamic stability."

3.         Evaluation of Fine Coordination and Proprioception:

Problem: The statement about fine coordination and proprioception is vague.

Improvement: Clearly state the findings related to fine coordination and proprioception. For example: "Contrary to expectations, no significant impairments were observed in the fine coordination and proprioception tests for the injured leg."

4.         Effectiveness of the Bandage:

Problem: The effectiveness of the bandage is not well-articulated.

Improvement: Clearly state the impact of the bandage. For example: "Subjects reported moderate to strong improvements in ankle stability and pain relief while wearing the bandage. The bandage significantly improved single leg stance performance, normalized single stance phase duration, and increased vertical ground reaction force during walking."

5.         Impact on Fine Coordination and Proprioception:

Problem: The impact of the bandage on fine coordination and proprioception is not clearly stated.

           Improvement: Clearly state the lack of impact on these areas. For example: "However, wearing the bandage did not produce significant improvements in fine coordination or proprioception."

6.         Conclusion Statement:

Problem: The conclusion statement is not strong and definitive.

Improvement: Make a definitive conclusion based on the findings. For example: "In conclusion, wearing a bandage during the acute phase of an ankle sprain significantly enhances motor performance, particularly in standing and walking tasks, despite having no marked effect on fine coordination and proprioception."

7.         Clinical Implications:

Problem: The clinical implications of the findings are not discussed.

Improvement: Discuss the clinical implications. For example: "These findings suggest that ankle bandages can be a valuable adjunct in the early rehabilitation phase to improve motor performance and reduce pain, potentially preventing further complications and enhancing recovery."

Comments on the Quality of English Language

Language Improvements:

Introduction:

Current: "The ankle is a complex joint with three degrees of freedom that enables the body to adapt to different surfaces during physical activities and to absorb shocks and forces."

Improvement: "The ankle is a complex joint with three degrees of freedom, enabling the body to adapt to various surfaces during physical activities and absorb shocks and forces."

Objective and Hypothesis:

Current: "The aim of this study was to investigate the effects of wearing an ankle bandage on fine coordination and proprioception as well as on motor performance in subjects 5 weeks after orthotic treatment for acute ankle injuries."

Improvement: "This study aimed to investigate the effects of wearing an ankle bandage on fine coordination, proprioception, and motor performance in subjects five weeks post-orthotic treatment for acute ankle injuries."

Methods:

Current: "All biomechanical investigations were conducted for subjects injured leg and the healthy leg with and without bandage in a randomized order, to minimize the influence of fatigue and habituation."

Improvement: "All biomechanical investigations were conducted on subjects' injured and healthy legs, with and without a bandage, in a randomized order to minimize the influence of fatigue and habituation."

Statistical Analysis:

Current: "Mean and standard deviations (mean ± SD) were calculated for all biomechanical variables."

Improvement: "Means and standard deviations (mean ± SD) were calculated for all biomechanical variables."

Results:

Current: "We found no statistically significant differences between any testing condition for the fine coordination and proprioception test."

Improvement: "No statistically significant differences were found between testing conditions for the fine coordination and proprioception test."

Discussion:

Current: "Our results might possibly be attributed to increased awareness and caution when moving the injured leg."

Improvement: "These results may be attributed to increased awareness and caution when moving the injured leg."

Conclusion:

Current: "We conclude that wearing a bandage in the acute phase of an ankle sprain may immediately improve motor performance, including standing and walking."

Improvement: "In conclusion, wearing a bandage during the acute phase of an ankle sprain may immediately improve motor performance, including standing and walking."

Specific Language Issues to Address:

Consistency:

Ensure consistent use of terms (e.g., "ankle bandage" vs. "bandage").

Maintain consistent verb tense throughout the sections.

Clarity and Precision:

Avoid vague terms and provide specific details where necessary.

Use precise language to describe methods, results, and interpretations.

Active vs. Passive Voice:

Prefer active voice for clarity and engagement. For example, "Researchers conducted tests" instead of "Tests were conducted by researchers."

Technical Terms:

Clearly define technical terms and ensure they are used correctly throughout the text.

Sentence Structure:

Vary sentence structure to improve readability. Avoid overly long or complex sentences that may confuse readers.

Grammar and Punctuation:

Check for correct use of grammar and punctuation, especially in complex sentences and lists.

Formal Tone:

Maintain a formal academic tone throughout the article. Avoid colloquial language and ensure professional phrasing.

Author Response

Response to Reviewer 1

The authors would like to thank you for your detailed review and for providing comments and suggestions to improve the quality of the manuscript. Reviewer comments and feedback have been incorporated, and the manuscript has been revised.

Open Review

Quality of English Language

( ) I am not qualified to assess the quality of English in this paper
( ) English very difficult to understand/incomprehensible
( ) Extensive editing of English language required
(x) Moderate editing of English language required
( ) Minor editing of English language required
( ) English language fine. No issues detected

Yes

Can be improved

Must be improved

Not applicable

Does the introduction provide sufficient background and include all relevant references?

( )

(x)

( )

( )

Is the research design appropriate?

( )

(x)

( )

( )

Are the methods adequately described?

( )

(x)

( )

( )

Are the results clearly presented?

( )

(x)

( )

( )

Are the conclusions supported by the results?

( )

(x)

( )

( )

Comments and Suggestions for Authors

For Title and Abstract:

  1. Clarity and Focus:

Problem: The introduction lacks a clear definition of the research problem and the hypothesis being tested.

Improvement: Clearly state the research question and the hypothesis in the abstract. For example: "This study investigates the hypothesis that wearing an ankle bandage immediately after an acute ankle injury improves fine coordination, proprioception, and motor performance."

Response: We thank you for your comment and your suggestion. We agree with the suggestion and have made some changes to the text.

  1. Specificity and Detail:

Problem: The abstract is vague about the specifics of the tests conducted and the results obtained.

Improvement: Provide more details on the methodology and results. For example: "Biomechanical investigations included fine coordination and proprioception tests, single leg stances, the Y-Balance test, and gait analysis. Results indicated moderate to strong improvements in ankle stability and pain relief while wearing the bandage."

Response: We agree with your comment. We have revised this paragraph.

  1. Consistency:

Problem: The text inconsistently references the timing of the intervention.

Improvement: Ensure consistent terminology regarding the timing. For example: "Subjects were tested five weeks post-injury to assess the immediate effects of the ankle bandage."

Response: The manuscript was reviewed by a native speaker to correct and maintain consistency in the tense.

  1. Impact of Findings:

Problem: The significance of the findings is not clearly articulated.

Improvement: Emphasize the implications of the findings for clinical practice. For example: "These findings suggest that ankle bandages may play a crucial role in early-stage rehabilitation by enhancing motor performance and reducing pain."

Response: We agree with your comment. This sentence was added to the text.

  1. Comparative Analysis:

Problem: There is no comparative analysis with existing literature.

Improvement: Compare your findings with previous studies. For example: "Unlike previous studies that focused on long-term effects, our study highlights the immediate benefits of ankle bandages in the acute phase of injury."

Response: We agree with your comment. However, the abstract should not exceed 200 words. Therefore, we have decided not to include this comparison in the abstract.

  1. Language Precision:

Problem: Some sentences are unclear or awkwardly phrased.

Improvement: Use precise language and revise awkward sentences. For example, revise "All subjects completed several rating questionnaires and biomechanical investigations" to "Subjects completed rating questionnaires and underwent comprehensive biomechanical assessments."

Response: We thank the reviewer for this comment and his suggestion. We agree with the suggestion and have revised this sentence.  Additionally, as mentioned above, the paper was reviewed by a native speaker to improve overall grammar and readability.

  1. Statistical Significance:

Problem: The abstract does not mention the statistical significance of the results.

Improvement: Include information about the statistical significance of the findings. For example: "Wearing the bandage significantly normalized single leg stance performance (p<0.05), stance phase duration (p<0.05), and vertical ground reaction forces during walking (p<0.05)."

Response: Sentence was revised und p-values were added.

  1. Scope and Limitations:

Problem: The abstract does not address the limitations of the study.

Improvement: Briefly mention the limitations. For example: "Further research is needed to explore the long-term effects and to confirm these findings in larger, more diverse populations.

Response:  We agree with your comment. But as mentioned above, the Abstract should not be longer than 200 words. So, we have decided not to include the limitations in the abstract.

For Introduction

Clarity and Focus:

Problem: The introduction lacks a clear, concise statement of the research problem and objective.

Improvement: Explicitly state the research problem and objective at the beginning. For example: "This study addresses the gap in research on the immediate effects of ankle bandages during the acute phase of ankle sprains, specifically focusing on their impact on fine coordination, proprioception, and motor performance."

Response: Thank you for your comment. However, we think it might be more suitable to mention the aims of the study towards the end of the introduction so that the reader clearly understands the background and why the study was conducted. As you suggested in your next comment, we started our introduction with the anatomy, types of injuries, and their classification, the incidence and prevalence (to highlight the importance of this study), followed by the treatment options and their limitations. We will then state our goal, why it might be beneficial to wear a bandage right after the acute phase of ankle sprains.

Logical Flow:

Problem: The introduction has a disjointed flow, making it difficult to follow the progression of ideas.

Improvement: Organize the information logically, ensuring a clear transition between sections. For example, start with the anatomy and function of the ankle, move to the types and consequences of ankle sprains, and then discuss the treatment and the current research gap.

Response: Thank you for your comment. In our introduction, we already tried to follow a logical sequence similar to the one you suggested. As described above, we begin our introduction with the anatomy, types of injuries, and their classification, the incidence and prevalence (to highlight the importance of this study), followed by the treatment options and their limitations. We then state our goal, why it might be beneficial to wear a bandage after the acute phase of ankle sprains. Nevertheless, to improve understanding we reorganized some sections and we have indented individual sections. Additionally, we have improved the transitions between some sections.

Specificity and Detail:

Problem: Some parts of the introduction are vague or overly broad.

Improvement: Provide specific details where necessary. For example: "Ankle sprains mostly occur following uncontrolled, sudden movements, such as plantarflexion and inversion, leading to excessive force or strain on the lateral ligaments of the joint."

Response: Thank you for your comment. To make the context of this study easier to understand for non-specialists, we decided to write from general to specific. In places where specific information was not necessary, we wrote more generally. In sections where more specific information was needed, as in the example sentence, we provided more detailed information. We believe that writing even more specifically could lead to a loss of understanding for the reader. Nevertheless, we have reviewed the introduction again for informational content.

Redundancy:

Problem: The introduction contains redundant information.

Improvement: Eliminate redundant sentences. For example, merge the sentences discussing the classification of ankle sprains by severity and duration to avoid repetition.

Response: Thank you for your comment. We checked for redundancy and reorganized and merged some sections.

Comparative Analysis:

Problem: There is a lack of comparative analysis with previous studies.

Improvement: Compare and contrast your study with previous research more explicitly. For example: "Unlike previous studies that have focused on the effects of bandages in healthy subjects or those with chronic ankle instability, our study examines their immediate impact during the acute phase of injury."

Response: We agree with your comment. This sentence was added to the text.

Language Precision:

Problem: Some sentences are unclear or awkwardly phrased.

Improvement: Use precise and clear language. For example, revise "During running and jumping, those forces can reach multiple times the body weight" to "During running and jumping, the forces exerted on the ankle joint can exceed several times the individual's body weight."

Response: We agree with your comment. This sentence was added to the text. To improve overall grammar and readability, the manuscript was reviewed again by a native speaker.

Significance of the Study:

Problem: The significance of the study is not clearly articulated.

Improvement: Clearly state the significance of your research. For example: "Understanding the immediate effects of ankle bandages can inform early intervention strategies, potentially reducing the incidence of chronic ankle instability and improving patient outcomes."

Response: We agree with your comment. This sentence was added to the text.

Statistical Data:

Problem: The introduction provides statistics without sufficient context or explanation.

Improvement: Integrate statistical data with explanatory context. For example: "Ankle sprains account for approximately 49% of all injuries in recreational athletes, highlighting the need for effective early treatment strategies."

Response: Thank you for your comment: The provided statistical information, in our opinion, fits the context and relevance of our study.

Hypothesis:

Problem: The hypothesis is mentioned but not clearly integrated into the introduction.

Improvement: Clearly state the hypothesis in the context of the research gap. For example: "We hypothesize that wearing an ankle bandage immediately after an acute lateral ankle sprain will reduce pain, enhance fine coordination and proprioception, and improve motor performance, including balance and gait."

Response: We agree with your comment. This sentence was added to the text.

For Materials and Methods

  1. Participant Description and Inclusion Criteria:

Problem: The inclusion and exclusion criteria are not clearly and comprehensively stated.

Improvement: Clearly define the inclusion and exclusion criteria with more detail and structure. For example: "Inclusion criteria were adults aged 18-60 years presenting with an acute unilateral supination ankle sprain (minimum grade 2) within three days post-injury. Exclusion criteria included grade 1 ankle sprains, upper leg sprains within the last 12 months, acute concomitant osseous injuries, history of confirmed lateral ligament injury, chronic ankle instability (CAI), neurological dysfunctions, rheumatic diseases, gout, arthrosis, recent surgeries, use of anticoagulants or corticosteroids, and any other conditions affecting motor performance or proprioception."

Response: Thank you for this comment and suggestion. We agree with the suggestion and have revised the text.

  1. Experimental Setup and Randomization:

Problem: The description of the experimental setup lacks detail on randomization and control measures.

Improvement: Specify how randomization was implemented and how control measures were maintained. For example: "All biomechanical tests were conducted on both the injured and healthy legs in a randomized order to minimize the influence of fatigue and habituation. Subjects performed familiarization trials with the bandage and tasks before the main trials."

Response: The text has been revised as suggested.

  1. Detailed Procedures for Biomechanical Tests:

Problem: The procedures for the biomechanical tests are not described with sufficient clarity and precision.

Improvement: Provide detailed and precise descriptions of each test. For example:

Fine Coordination and Proprioception Test: "A customized foot pedal allowing inversion and eversion movements within a 5° range was used. Subjects were instructed to trace a target sine wave displayed on a screen by manipulating the pedal. Five trials of 18 seconds each were performed, with the mean deviation from the target sine wave calculated for the last two trials within the 6-16 second interval."

Single Leg Stance: "Subjects stood on a pressure distribution platform, maintaining an upright posture with their knee straight but not locked, arms crossed in front of the chest, and gaze directed ahead. Each trial lasted 20 seconds, with center of pressure (COP) parameters measured for analysis."

Y-Balance Test: "Subjects performed the Y-Balance test standing on one foot and reaching with the other in three directions (anterior, posteromedial, posterolateral). Maximum reach distances were normalized to leg length, and any loss of balance or stance foot movement necessitated a trial repeat."

Gait Analysis: "Subjects walked ten times over a pressure distribution platform integrated into a walkway. Spatial-temporal gait parameters, including gait velocity, step length, and vertical ground reaction forces, were recorded."

Response: We thank the reviewer for the comments and his suggestions. We have revised the method section.

  1. Subjective Ratings:

Problem: The method for collecting subjective ratings is not clearly described.

Improvement: Clearly describe the process and tools used for subjective ratings. For example: "Subjects provided anthropometric data and rated their pain and stability improvements while walking and standing barefoot, in shoes, and with the bandage using visual analogue scales (0-10)."

Response: This part of the methods has also been revised.

  1. Statistical Analysis:

Problem: The statistical analysis section is brief and lacks detail.

Improvement: Expand the statistical analysis section to include more details on the methods used. For example: "Mean and standard deviations were calculated for all biomechanical variables. The Shapiro-Wilk test was used to assess normality. A one-way ANOVA for repeated measures followed by Bonferroni post-hoc tests was conducted to evaluate differences between conditions (healthy leg, injured leg with and without bandage). Statistical significance was set at α = 0.05. Effect sizes were calculated using Cohen's d, categorized as trivial (< 0.2), small (< 0.5), medium (< 0.8), or large (≥ 0.8)."

Response: The text has been revised as suggested.

  1. Clarity and Precision:

Problem: Some sentences are unclear or awkwardly phrased.

Improvement: Use precise and clear language throughout. For example, revise "The bandage, which was used in this study (MalleoTrain® Bauerfeind AG), consisted of an elastic but tight-fitting high-low knit" to "The MalleoTrain® bandage (Bauerfeind AG) used in this study is made of an elastic, tight-fitting high-low knit material, providing alternating pressure massage during movement."

Response: Thank you for this comment. In order to enhance readability and clarity, sentences that were found to be imprecise and unclear were revised.

For Results:

  1. Demographic and Clinical Data Presentation:

Problem: The presentation of demographic and clinical data is unclear and lacks detail.

Improvement: Present demographic and clinical data in a clear and detailed manner. For example: "Table 1 presents the demographic and clinical characteristics of the subjects. The cohort included 70 subjects with a slight predominance of males (n=36) over females (n=34). Most injuries were to the right leg (n=38), with the remainder affecting the left leg (n=32). The anterior talofibular ligament was the most frequently affected structure, followed by the calcaneofibular and posterior talofibular ligaments. Commonly, injuries were associated with swelling and other general complaints."

Response: Thank you for your comments and suggestions. We have revised this section.

  1. Statistical Analysis and Interpretation:

Problem: The statistical analysis is not clearly explained, and the significance of the results is not fully interpreted.

Improvement: Clearly explain the statistical methods and interpret the results. For example: "The Shapiro-Wilk test confirmed the normal distribution of variables. One-way ANOVA for repeated measures followed by Bonferroni post-hoc tests identified statistically significant differences between conditions. Effect sizes (Cohen’s d) were calculated to quantify the magnitude of differences, with d ≥ 0.8 considered large."

Response: Those information belong to the methods section (2.3 Statistical Analysis), which has been revised and all relevant information have been included.

  1. Subjective Ratings:

Problem: The description of subjective pain ratings is vague.

Improvement: Provide more context and detail. For example: "Table 2 summarizes self-rated pain perceptions using a visual analogue scale (0-10). Subjects reported mild to moderate pain, which increased during walking compared to standing. The bandage generally alleviated pain. No significant differences in pain perception were noted between walking barefoot and in shoes."

Response: The text has been revised as suggested.

  1. Figures and Tables:

Problem: Figures and tables are referenced without proper integration into the text.

Improvement: Integrate figures and tables into the text with clear references. For example: "As shown in Figure 3, subjects reported various improvements due to the bandage, with strong stabilizing effects noted by half the cohort and moderate effects by the other half. Pain improvement was mostly moderate, with notable improvements in single leg stance and Y-Balance tests."

Response: The text has been revised.

  1. Results Interpretation:

Problem: The interpretation of results lacks depth and clarity.

Improvement: Provide a more detailed interpretation of the results. For example: "No statistically significant differences were observed in the fine coordination and proprioception test between conditions (p=0.078), although there was a trend towards smaller deviations with the bandage. Significant differences were found in COP length during single leg stance (p<0.001), indicating improved stability with the bandage."

Response: The text has undergone a revision. However, statements like “indicating improved stability with the bandage” are an interpretation of the results and therefore belong to the discussion section.

  1. Data Consistency:

Problem: Inconsistent data presentation and lack of detail in some areas.

Improvement: Ensure consistency and completeness in data presentation. For example, consistently report means ± standard deviations and provide comprehensive details in tables: "Table 3 presents the mean deviation in degrees from the target sine wave across conditions. While no significant differences were detected, the trend suggests potential benefits of the bandage."

Response: The text has been revised.

  1. Use of Technical Language:

Problem: Some technical terms and statistical measures are not well explained.

Improvement: Clearly explain technical terms and statistical measures. For example: "COP (center of pressure) parameters, including COP length and 95% confidence area, were used to assess postural stability. Longer COP excursions indicate greater instability, while smaller confidence areas suggest improved stability."

Response: This information has been added to the methods section.

  1. Comprehensive Reporting:

Problem: The results section lacks comprehensive reporting on some tests.

Improvement: Provide a complete report on all tests conducted. For example: "The Y-Balance test revealed significantly higher reach distances on the healthy leg compared to the injured leg with or without the bandage, particularly in the anterior direction (Figure 4). Gait analysis showed no significant differences in gait velocity and double stance phase between conditions (Table 5)."

Response: A summary of the results can be found in the conclusion.

For Discussion:

  1. Objective and Hypothesis Clarity:

Problem: The aim and hypothesis are not stated clearly and concisely.

Improvement: Clearly and concisely state the aim and hypothesis. For example: "This study aimed to investigate the effects of wearing an ankle bandage on fine coordination, proprioception, and motor performance in subjects five weeks post-orthotic treatment for acute ankle injuries. We hypothesized that the bandage would reduce pain, improve fine coordination and proprioception, and enhance motor performance, including static and dynamic balance as well as gait normalization."

Response: Thank you for your comment. We agree with the suggestion and have made some changes to the text.

  1. Section Organization:

Problem: The sections are not well-organized, making it difficult to follow the results and discussion.

Improvement: Clearly separate and organize sections for better readability. For example:

  • 4.1. Fine Coordination and Proprioception Test
  • 4.1.1. Effects of the Injury
  • 4.1.2. Effects of the Bandage
  • 4.2. Motor Performance
  • 4.2.1. Effects of the Injury
  • 4.2.2. Effects of the Bandage

Response: Thank you for your comment. In fact, our structure in the discussion section already aligns with the suggested structure. Nevertheless, we have adjusted the structure of the results to better match that of the discussion. This should enhance the readability, comprehensibility, and coherence between the results and the discussion.

  1. Statistical Results Reporting:

Problem: The statistical results are not reported with enough detail or clarity.

Improvement: Clearly report the statistical results, including p-values and effect sizes where applicable. For example: "No statistically significant differences were found for the fine coordination and proprioception test between conditions (p > 0.05), although there was a trend towards better performance with the bandage (d = 0.2)."

Response: Thank you for your comment. To maintain readability and because several test parameters showed significant differences, which would require including numerous p-values and effect size values in the text, we opted against including them in the main text. Moreover, these values are already organized and comparably presented in the results section, thus avoiding redundancy.

  1. Comparison to Existing Literature:

Problem: The comparison to existing literature is vague and lacks detail.

Improvement: Provide specific comparisons to existing literature. For example: "Our findings contrast with studies on chronic ankle instability (CAI) which typically show impaired proprioceptive abilities post-injury [3,13-18,29,51,52]. This discrepancy may be due to the acute phase of injury in our study, where heightened awareness and active healing processes may play a role."

Response: Thank you for your comment. When writing the paper, we have indeed tried to explain possible differences between our results and those of other studies and to discuss the existing literature in this context. Where appropriate and beneficial for understanding, we have also provided more detailed discussions. However, to avoid overloading the already extensive discussion, we have refrained from unnecessary information. Nevertheless, we have made some changes to the text.

  1. Mechanisms and Theoretical Explanations:

Problem: The mechanisms and theoretical explanations are not well-developed.

Improvement: Provide detailed theoretical explanations for the findings. For example: "The trend towards better performance with the bandage may be due to increased mechanical stability and proprioceptive feedback from cutaneous mechanoreceptors [3,26,53,59,60]. The bandage may also provide compression, stimulating underlying structures and enhancing proprioceptive acuity."

Response: Thank you for your comment. Due to the closely interconnected and interdependent arguments, it appears challenging to adopt your suggestion without completely restructuring significant parts of the text. However, we have made some changes to the text. We will also take into account your valuable feedback for the upcoming paper that we are currently planning.

  1. Subjective Data Integration:

Problem: Subjective data is mentioned but not integrated well into the discussion.

Improvement: Integrate subjective data into the discussion. For example: "Subjective reports indicated that 50% of subjects experienced strong stabilizing effects with the bandage, which aligns with the observed trends towards improved proprioceptive performance."

Response: Thank you for your comment. We appreciate your suggestion and have revised the text accordingly.

  1. Discussion of Motor Performance:

Problem: The discussion on motor performance is fragmented and lacks depth.

Improvement: Discuss motor performance results in depth, linking them to theoretical frameworks. For example: "Motor performance impairments in the injured leg, such as increased sway and reduced reach distance, may be attributed to mechanical ankle instability (MAI) and functional ankle instability (FAI), affecting proprioceptive and neuromuscular control [29,48,65,66]."

Response: Thank you for your comment. Indeed, we have already tried to discuss motor performance extensively based on numerous other studies and arguments. Each of the three motor performance tests has been carefully studied in the literature to understand how injury and bandaging affect them. Adding more to this discussion might make the text harder to understand, as it's already quite long. However, we have made some changes to the text. Also, we'll consider your helpful feedback for our next paper that we're planning now.

  1. Gait Analysis Clarity:

Problem: The gait analysis results are not clearly explained.

Improvement: Provide a clear explanation of gait analysis results. For example: "Gait analysis revealed longer steps and shorter single stance phases for the injured leg, suggesting compensatory mechanisms to minimize load on the injured leg. This aligns with findings of reduced ground reaction forces in the loading and terminal stance phases [54,83-85]."

Response: Thank you for your comment. Due to the closely interconnected and interdependent arguments, it seems challenging to accept your suggestion without completely restructuring significant parts of the text. Since most gait parameters exhibit dependencies among each other and changing one parameter affects another, we have chosen this approach. Nonetheless, we have made some changes to the text. We will also consider your valuable feedback for the upcoming paper that we are currently planning.

  1. Limitations and Future Research:

Problem: Limitations and suggestions for future research are not well-articulated.

Improvement: Clearly state the limitations and future research directions. For example: "Limitations of this study include the absence of MRI data to precisely identify injury types and the lack of kinematic and electromyographic analyses. Future research should explore these aspects and compare the efficacy of different types of ankle supports."

Response: Thank you for your comment. We appreciate your suggestion and have revised the text accordingly.

  1. Overall Language Precision:

Problem: The language is sometimes unclear and imprecise.

Improvement: Use precise and clear language throughout. For example, revise "These results were somewhat surprising since various other studies have shown impaired proprioceptive abilities" to "Contrary to expectations, our results showed no significant proprioceptive impairments, unlike previous studies on CAI [3,13-18,29,51,52]."

Response: Thank you for your comment. We appreciate your suggestion and have revised the text accordingly. To improve overall grammar and readability, the manuscript was reviewed again by a native speaker. Corresponding changes have been made in the text.

Conclusion:

  1. Summary of Findings:

Problem: The summary of findings is not concise and lacks clarity.

Improvement: Clearly and concisely summarize the key findings. For example: "Five weeks post-ankle injury, subjects exhibited mild to moderate pain and ankle instability, with impaired motor performance in the injured leg compared to the healthy leg. This was evident from higher sway values during single leg stances, reduced reach distances in the Y-Balance test, and altered gait characterized by longer steps, shorter stance phases, and reduced vertical ground reaction forces."

Response: Thank you for your comment. We appreciate your suggestion and have revised the text accordingly.

  1. Details on Impairments:

Problem: The description of impairments lacks detail and specificity.

Improvement: Provide specific details about the impairments. For example: "Motor performance impairments included significant increases in postural sway during single leg stances and a notable reduction in reach distances in the Y-Balance test, indicating decreased dynamic stability."

Response: Thank you for your comment. We have made some changes to the text.

  1. Evaluation of Fine Coordination and Proprioception:

Problem: The statement about fine coordination and proprioception is vague.

Improvement: Clearly state the findings related to fine coordination and proprioception. For example: "Contrary to expectations, no significant impairments were observed in the fine coordination and proprioception tests for the injured leg."

Response: Thank you for your comment. We appreciate your suggestion and have revised the text accordingly.

  1. Effectiveness of the Bandage:

Problem: The effectiveness of the bandage is not well-articulated.

Improvement: Clearly state the impact of the bandage. For example: "Subjects reported moderate to strong improvements in ankle stability and pain relief while wearing the bandage. The bandage significantly improved single leg stance performance, normalized single stance phase duration, and increased vertical ground reaction force during walking."

Response: Thank you for your comment. We appreciate your suggestion and have revised the text accordingly.

  1. Impact on Fine Coordination and Proprioception:

Problem: The impact of the bandage on fine coordination and proprioception is not clearly stated.

  • Improvement: Clearly state the lack of impact on these areas. For example: "However, wearing the bandage did not produce significant improvements in fine coordination or proprioception."

Response: Thank you for your comment. We appreciate your suggestion and have revised the text accordingly.

  1. Conclusion Statement:

Problem: The conclusion statement is not strong and definitive.

Improvement: Make a definitive conclusion based on the findings. For example: "In conclusion, wearing a bandage during the acute phase of an ankle sprain significantly enhances motor performance, particularly in standing and walking tasks, despite having no marked effect on fine coordination and proprioception."

Response: Thank you for your comment. We appreciate your suggestion and have revised the text accordingly.

  1. Clinical Implications:

Problem: The clinical implications of the findings are not discussed.

Improvement: Discuss the clinical implications. For example: "These findings suggest that ankle bandages can be a valuable adjunct in the early rehabilitation phase to improve motor performance and reduce pain, potentially preventing further complications and enhancing recovery."

Response: Thank you for your comment. We appreciate your suggestion and have revised the text accordingly.

Comments on the Quality of English Language

Language Improvements:

Introduction:

Current: "The ankle is a complex joint with three degrees of freedom that enables the body to adapt to different surfaces during physical activities and to absorb shocks and forces."

Improvement: "The ankle is a complex joint with three degrees of freedom, enabling the body to adapt to various surfaces during physical activities and absorb shocks and forces."

Response: Thank you for your suggestions. To improve overall grammar and readability, the manuscript was reviewed again by a native speaker.

Objective and Hypothesis:

Current: "The aim of this study was to investigate the effects of wearing an ankle bandage on fine coordination and proprioception as well as on motor performance in subjects 5 weeks after orthotic treatment for acute ankle injuries."

Improvement: "This study aimed to investigate the effects of wearing an ankle bandage on fine coordination, proprioception, and motor performance in subjects five weeks post-orthotic treatment for acute ankle injuries."

Response: Thank you for your suggestions. The text has been revised.

 Methods:

Current: "All biomechanical investigations were conducted for subjects injured leg and the healthy leg with and without bandage in a randomized order, to minimize the influence of fatigue and habituation."

Improvement: "All biomechanical investigations were conducted on subjects' injured and healthy legs, with and without a bandage, in a randomized order to minimize the influence of fatigue and habituation."

 Response: Thank you for your suggestions. The text has been revised.

Statistical Analysis:

Current: "Mean and standard deviations (mean ± SD) were calculated for all biomechanical variables."

Improvement: "Means and standard deviations (mean ± SD) were calculated for all biomechanical variables."

 Response: Thank you for your suggestions. The text has been revised.

Results:

Current: "We found no statistically significant differences between any testing condition for the fine coordination and proprioception test."

Improvement: "No statistically significant differences were found between testing conditions for the fine coordination and proprioception test."

Response: Thank you for your suggestions. The text has been revised.

Discussion:

Current: "Our results might possibly be attributed to increased awareness and caution when moving the injured leg."

Improvement: "These results may be attributed to increased awareness and caution when moving the injured leg."

Response: Thank you for your suggestions. The text has been revised.

Conclusion:

Current: "We conclude that wearing a bandage in the acute phase of an ankle sprain may immediately improve motor performance, including standing and walking."

Improvement: "In conclusion, wearing a bandage during the acute phase of an ankle sprain may immediately improve motor performance, including standing and walking."

Response: Thank you for your suggestions. The text has been revised.

Specific Language Issues to Address:

Consistency:

Ensure consistent use of terms (e.g., "ankle bandage" vs. "bandage").

Maintain consistent verb tense throughout the sections.

Response: We have checked and improved it, but in some places, 'ankle bandage' must be mentioned.

Clarity and Precision:

Avoid vague terms and provide specific details where necessary.

Use precise language to describe methods, results, and interpretations.

Response: We have checked this and improved the manuscript accordingly.

Active vs. Passive Voice:

Prefer active voice for clarity and engagement. For example, "Researchers conducted tests" instead of "Tests were conducted by researchers."

Response: We have checked this and improved the manuscript accordingly.

Technical Terms:

Clearly define technical terms and ensure they are used correctly throughout the text.

Response: We have checked this and improved the manuscript accordingly.

Sentence Structure:

Vary sentence structure to improve readability. Avoid overly long or complex sentences that may confuse readers.

 Response: We have checked this and improved the manuscript accordingly.

Grammar and Punctuation:

Check for correct use of grammar and punctuation, especially in complex sentences and lists.

 Response: We have checked this and improved the manuscript accordingly.

Formal Tone:

Maintain a formal academic tone throughout the article. Avoid colloquial language and ensure professional phrasing.

Response: We have checked this and improved the manuscript accordingly.

Submission Date

29 May 2024

Date of this review

04 Jun 2024 13:33:51

Many thanks again for your comments and constructive criticism. We hope the revised manuscript conveys the content more precisely and corresponds with your expectations.

Sincerely,

Tobias Heß

Reviewer 2 Report

Comments and Suggestions for Authors

The authors investigate the impact of ankle bandages on early rehabilitation five weeks post-ankle sprain. They tested 70 subjects with acute unilateral supination trauma using rating questionnaires and biomechanical assessments, including coordination and proprioception tests, single-leg stances, the Y-Balance test, and gait analysis. Subjects reported moderate to strong improvements in ankle stability and pain with the bandage. The bandage significantly improved single-leg stance performance, stance phase duration, and vertical ground reaction forces during walking. However, it did not significantly affect fine coordination and proprioception. The authors concluded that wearing a bandage in the acute phase of an ankle sprain can immediately enhance motor performance, particularly in standing and walking.

The article is overall interesting and well-written, aiming to improve the rehabilitative treatment of acute ankle sprain. However, some methodological aspects must be clarified to extend the validity of the conclusions to all patients suffering from ankle sprain.

Title: please change “ankle injury” with “ankle sprain”

Abstract and keywords: Ok.

Introduction: OK. Aim of the study and hypothesis are specified.

Materials and methods:

1)      Specify the nature of the study (retrospective, prospective single cohort…)

2)      The study must specify and quantify the source population and the selection process (possibly using a flow-chart). The study must specify if subjects included Subjects are representative of:

a.       the entire source population

b.       an unselected sample of consecutive patients

c.       a random sample (specify randomization methods).

Indicate the size of the source population and the recruitment period. Report the proportion of subjects who agreed to participate and those who refused. To validate representativeness and generalize findings, show that key confounding factors (age, sex, BMI, etc.) in the study sample match those of the source population.

3)      Specify the sample size estimation based on your hypothesis and primary expected outcome.

Results: OK.

Discussion: Where possible, please condense the discussion and highlight the essential points within 1-2 pages to make it more engaging and easier to read.

Conclusion and references: OK.

Author Response

Response to Reviewer 2

The authors would like to thank you for your detailed review and for providing comments and suggestions to improve the quality of the manuscript. Reviewer comments and feedback have been incorporated, and the manuscript has been revised.

Open Review

Quality of English Language

( ) I am not qualified to assess the quality of English in this paper
( ) English very difficult to understand/incomprehensible
( ) Extensive editing of English language required
( ) Moderate editing of English language required
( ) Minor editing of English language required
(x) English language fine. No issues detected

Yes

Can be improved

Must be improved

Not applicable

Does the introduction provide sufficient background and include all relevant references?

(x)

( )

( )

( )

Is the research design appropriate?

( )

(x)

( )

( )

Are the methods adequately described?

( )

(x)

( )

( )

Are the results clearly presented?

(x)

( )

( )

( )

Are the conclusions supported by the results?

(x)

( )

( )

( )

Comments and Suggestions for Authors

The authors investigate the impact of ankle bandages on early rehabilitation five weeks post-ankle sprain. They tested 70 subjects with acute unilateral supination trauma using rating questionnaires and biomechanical assessments, including coordination and proprioception tests, single-leg stances, the Y-Balance test, and gait analysis. Subjects reported moderate to strong improvements in ankle stability and pain with the bandage. The bandage significantly improved single-leg stance performance, stance phase duration, and vertical ground reaction forces during walking. However, it did not significantly affect fine coordination and proprioception. The authors concluded that wearing a bandage in the acute phase of an ankle sprain can immediately enhance motor performance, particularly in standing and walking.

The article is overall interesting and well-written, aiming to improve the rehabilitative treatment of acute ankle sprain. However, some methodological aspects must be clarified to extend the validity of the conclusions to all patients suffering from ankle sprain.

Title: please change “ankle injury” with “ankle sprain”

Response: Thank you for your comment. We replaced “ankle injury” with “ankle sprain” in the title and at various other locations in the manuscript.  

Abstract and keywords: Ok.

Response: Thank you. 

Introduction: OK. Aim of the study and hypothesis are specified.

Response: Thank you.

Materials and methods:

1)      Specify the nature of the study (retrospective, prospective single cohort…)

Response: Thank you for your comment. We added the information “interventional study” to the materials and methods section.

2)      The study must specify and quantify the source population and the selection process (possibly using a flow-chart). The study must specify if subjects included Subjects are representative of:

  1. the entire source population
  2. an unselected sample of consecutive patients
  3. a random sample (specify randomization methods).

Indicate the size of the source population and the recruitment period. Report the proportion of subjects who agreed to participate and those who refused. To validate representativeness and generalize findings, show that key confounding factors (age, sex, BMI, etc.) in the study sample match those of the source population.

Response: Thank you for your comment. We added the following sentence to the methods section: “Subjects were recruited independently of their ethnicity and the cause of injury.” Information about the inclusion and exclusion criteria has already been mentioned in section “2.1. Participants”. Table 1 also provides further details on relevant anthropometric data, as well as the total number, gender distribution, and injured structures.

3)      Specify the sample size estimation based on your hypothesis and primary expected outcome.

Response: Thank you for your comment. The sample size planning is based on a G-Power analysis using data from a pilot study that we conducted prior to this main study.

Results: OK.

Response: Thank you.

Discussion: Where possible, please condense the discussion and highlight the essential points within 1-2 pages to make it more engaging and easier to read.

Response: Thank you for your comment. We acknowledge that our manuscript is lengthy. However, the length can be explained by the fact that it contains multiple focal points, each of which is addressed separately. Consequently, we investigated and discussed the effects of the ankle sprain and the effect of the ankle bandage on three distinct motor performance tests, as well as on fine coordination and proprioception. Most of these topics are related to different mechanisms and therefore need to be discussed separately. To aid reader comprehension, we used text indentations and maintained a consistent content sequence for each topic throughout the discussion. Readers seeking a more in-depth argumentation can refer to the discussion, or alternatively, obtain a quicker overview by reading the abstract or conclusion. Nevertheless, we will consider your valuable advice for the next paper we are currently planning.

Conclusion and references: OK.

Response: Thank you

Many thanks again for your comments and constructive criticism. We hope the revised manuscript conveys the content more precisely and corresponds with your expectations.

Sincerely,

Tobias Heß

Round 2

Reviewer 1 Report

Comments and Suggestions for Authors

Dear Authors,

Thank you for your thorough and diligent revisions in response to the feedback provided. I have carefully reviewed the updated manuscript and am pleased to inform you that the quality of the article has significantly improved as a result of your efforts. The manuscript now exhibits a high level of clarity, coherence, and academic rigor, making it worthy of publication.

The title and abstract have been enhanced to clearly state the research question and hypothesis, providing a focused overview of the study. The methodological details and results are now better specified, and the inclusion of statistical significance values has strengthened the impact of your findings. The abstract effectively communicates the significance of the study, which is crucial for capturing the interest of the readers.

The introduction is now well-organized, logically flowing from the anatomy and function of the ankle to the types and consequences of ankle sprains, and finally to the research gap and study objectives. This logical progression helps the reader understand the context and importance of your research. Additionally, the introduction now includes specific details and comparative analyses with previous studies, which enhances its depth and relevance.

In the materials and methods section, the participant inclusion and exclusion criteria are clearly defined, and detailed descriptions of the experimental setup, randomization, and biomechanical tests provide a comprehensive understanding of the methodology. The statistical analysis section has been expanded to include detailed methods, which enhances the rigor of your study.

The results section now presents demographic and clinical data clearly and with sufficient detail. The interpretation of results is more thorough, with clear explanations of statistical findings. Subjective ratings are integrated into the overall results, providing a more holistic view of the study outcomes. The figures and tables are well-referenced and integrated into the text, which aids in the reader’s understanding.

The discussion is well-structured and logically organized, providing a detailed comparison to existing literature. The theoretical explanations and implications of the results are well-developed, and the limitations and future research directions are clearly articulated. This comprehensive discussion helps to contextualize your findings within the broader field of study.

The conclusion succinctly summarizes the key findings and their implications, making a definitive statement based on the results. The clinical implications are discussed, highlighting the potential impact of the study on early rehabilitation strategies.

Overall, the language throughout the manuscript is precise and clear, significantly improving readability and comprehension. Technical terms are clearly defined and consistently used, enhancing the manuscript's academic rigor. The formal tone is maintained throughout, ensuring a professional and scholarly presentation.

In summary, the revised manuscript successfully addresses the initial concerns and demonstrates significant improvement in quality. It is now a well-structured, clear, and academically rigorous article that is worthy of publication. Thank you for your hard work and responsiveness to my comments.

Sincerely,

Prof. Dr. Paul-Dan Sirbu